# Biological Activities of Ruthenium NHC Complexes: An Update

**DOI:** 10.3390/antibiotics12020365

**Published:** 2023-02-09

**Authors:** Alessia Catalano, Annaluisa Mariconda, Maria Stefania Sinicropi, Jessica Ceramella, Domenico Iacopetta, Carmela Saturnino, Pasquale Longo

**Affiliations:** 1Department of Pharmacy-Drug Sciences, University of Bari “Aldo Moro”, 70126 Bari, Italy; 2Department of Science, University of Basilicata, 85100 Potenza, Italy; 3Department of Pharmacy, Health and Nutritional Sciences, University of Calabria, 87036 Arcavacata di Rende, Italy; 4Department of Chemistry and Biology, University of Salerno, Via Giovanni Paolo II, 132, 84084 Fisciano, Italy

**Keywords:** Ru-NHC complexes, *N*-heterocyclic carbenes, antitumor agents, antiproliferative activity, antibacterials, antimicrobials, antifungals

## Abstract

Ruthenium *N*-heterocyclic carbene (NHC) complexes have unique physico-chemical properties as catalysts and a huge potential in medicinal chemistry and pharmacology, exhibiting a variety of notable biological activities. In this review, the most recent studies on ruthenium NHC complexes are summarized, focusing specifically on antimicrobial and antiproliferative activities. Ruthenium NHC complexes are generally active against Gram-positive bacteria, such as *Bacillus subtilis*, *Staphylococcus aureus, Micrococcus luteus*, *Listeria monocytogenes* and are seldom active against Gram-negative bacteria, including *Salmonella typhimurium*, *Pseudomonas aeruginosa* and *Escherichia coli* and fungal strains of *Candida albicans.* The antiproliferative activity was tested against cancer cell lines of human colon, breast, cervix, epidermis, liver and rat glioblastoma cell lines. Ruthenium NHC complexes generally demonstrated cytotoxicity higher than standard anticancer drugs. Further studies are needed to explore the mechanism of action of these interesting compounds.

## 1. Introduction

Ruthenium (II) complexes represent an important class of organometallic compounds with numerous applications in the fields of homogeneous, heterogeneous and photocatalytic catalysis [1,2], including biological stains, and in the field of therapy [3,4]. Ruthenium complexes endowed with anticancer properties [5] are attracting significant attention, due to their vast and different structural types and their ability to variously bind different ligands with the advantage of lower toxicity than platinum complexes [6,7,8]. These compounds reasonably penetrate more efficiently tumor cells and effectively bind to DNA, from which comes the suggestion of their use in different cancers [9,10], including lung [11], breast [12], ovarian [13,14,15,16] and colorectal ones [8]. Recently, nanomedicine formulations of metal complexes developed for the treatment of cancers are under study [17]. Moreover, numerous other biological activities [18], such as antioxidant, anti-inflammatory [19], and antimicrobial [3,20,21], have been described for ruthenium complexes. Recent studies are also addressed to the use of ruthenium complexes as antivirals for the treatment of COVID-19 [22]. Ruthenium complexes include those with Schiff bases [23,24,25], phosphines [26], carbazole [27], *N*-heterocyclic carbenes (NHCs), cycloruthenated and half-sandwiched compounds [28]. Considering the huge variety of ruthenium (II) complexes, this review will focus merely on one class of them, namely Ru-NHC complexes. Ru-NHC complexes have promising catalytic potential for a vast range of synthetic applications, including the activity in transfer hydrogenation [29,30] of ketones, metathesis reactions [31], secondary alcohol oxidation [32], *N*-alkylation of amines, amides, and sulfonamides [33,34] and Oppenauer-type oxidation [35,36]. The high catalytic activity of these compounds is even comparable to that of the Noels catalyst [37]. More importantly, Ru-NHC complexes exhibit notable pharmaceutical activities. The importance of these compounds is also well-established by recent computational studies [38], suggesting the structural basis of the biomolecular action of these compounds. In this review, we want to highlight the importance of the antimicrobial and antiproliferative activities of the Ru(II)-NHC complexes that may be considered significant starting points for the development of new antimicrobial and anticancer agents, providing an update of recent studies inherent this relevant topic in medicinal chemistry in the last five years.

## 2. *N*-Heterocyclic Carbenes (NHCs)

N-heterocyclic carbenes (NHCs) are a category of electron donor ligands able to form dative metal–ligand bonds, leading to a universal class of compounds in organometallic and coordination chemistry. NHCs typically mimic the chemical properties of phosphines [39]. They belong to five different families: imidazolinylidenes, imidazolylidenes, triazolinylidenes, thiazolilylidenes and pyrazolinylidenes (Figure 1) [40].

NHCs are suitable for efficient design because they can be synthesized within a few steps, offering the possibility of the N and C functionalization for structure modification. They act as excellent σ-donors, generating a structural variety ranging from linear, square pyramidal, trigonal bipyramidal, tetrahedral and octahedral geometries of metal-NHC complexes. The essential role of NHCs is related to their ability to form complexes with metals. Metal-NHCs are widely used for organic processes, such as the formation of amide linkage, hydrogenation, isomerization, cycloisomerization, cyclopropanation, hydrosilylation, allylation and deallylation, enol-ester synthesis, heterocycle synthesis and C-C alkyne coupling [40]. Most importantly, different organometallic-NHC complexes, such as silver, gold, platinum, copper, palladium and selenium demonstrated interesting biological properties [41,42,43,44], including antimicrobial [45,46,47], anticancer [48,49,50,51,52,53,54], antiparasitic [55], hemolytic and thrombolytic activities [56,57]. Recent studies for gold and silver-NHC compounds are addressed to breast [58,59], ovarian [60] and cervical human cancer [61]. Research has been carried out in order to understand the mechanism of action of gold and silver-NHC complexes as anticancer agents, finding activity against human topoisomerases I and II [62,63], actin [64] and tubulin [65], or triggering the reactive oxygen species-dependent intrinsic apoptotic pathway [66]. Recently, gold and silver compounds with NHC have been indicated as promising compounds for the treatment of COVID-19 [67], as they demonstrated a strong inhibition of the S/ACE2 interaction and particularly of the PL^pro^ enzymatic activity [68]. Moreover, an Ag-*N*-heterocyclic carbene complex bearing the hydroxyethyl ligand determined the inhibition of human carbonic anhydrase I (hCA I) and hCA II isoenzymes, α-glycosidase and AChE and BChE enzymes, thus suggesting the selection of NHC complexes for further studies in glaucoma, epilepsy and other diseases related to metabolic enzymes [69].

## 3. Ruthenium-NHC Complexes

Besides the importance of ruthenium-NHC complexes in organometallic catalysis [70,71] and bioinorganic chemistry, they have been described as effective anticancer and antimicrobial agents [40,43,47]. Patil et al., 2020, published the advances in the design, synthesis, characterization and biomedical applications, particularly the antimicrobial and anticancer activities, of ruthenium NHC–metal complexes and other metals (silver, gold, palladium, rhodium, iridium, and platinum), covering works published from 2015 to 2020 [39]. Several studies regard only the activity of Ru-NHC complexes as antiproliferative agents, whereas the antimicrobial activity is almost always studied along with other activities, including antiproliferative and antioxidant. The two paragraphs below summarized these studies.

### 3.1. Ru(II)-NHC Complexes with Antiproliferative Activity

Recent studies regarding the antiproliferative activity of Ru(II)-NHC complexes are summarized and the half-maximal (50%) inhibitory concentration (IC_50_) values are given, when reported in the literature.

Lam et al., 2018, [72] investigated several halide-substituted benzimidazolium-derived NHC of Ru^II^/Os^II^ complexes, using NHCs that were symmetrically and non-symmetrically methyl- and benzyl-substituted, and reported their inhibition of the selenoenzyme thioredoxin reductase (TrxR) and antiproliferative activities. The anticancer activity was studied against human colon (HCT-116), human cervical (SiHa) and human breast cancer (NCI-H460) cells. The diiodido(1,3-dibenzylbenzimidazol-2-ylidene)(*η*^6^-*p*-cymene)ruthenium(II) complex **1** is a potent TrxR inhibitor and an antiproliferative agent. The authors found out that there was no clear correlation between the two activities, thus, it was suggested that TrxR inhibition was unlikely to be the main mode of action.

Tabrizi et al., 2019, [73] studied the in vitro antiproliferative potential and cyclooxygenase-2 (COX-2) inhibitory activity of a cyclometalated Ru(II) complex (**2**) containing ibuprofen (Ibu), 1,3,5-triaza-7-phosphaadamantane (PTA) and a CCC-pincer containing naproxen moiety (CCC-Nap). Antiproliferative studies were carried out on breast cancer (MCF-7 and MDA-MB-231), colon cancer (HT-29) cell lines, healthy breast cell lines (MCF-10A), and human embryonic kidney normal cells (HEK293) by means of 3-(4,5-dimethylthiazol-2-yl)-2,5-diphenyltetrazolium bromide (MTT) assay, after 72 h exposure. The complex **2** was quite potent, about twice as active as cisplatin against breast cancer cells, and around 14 times less active against HT-29 cell lines than cisplatin. Interestingly, the complex **2** inhibition studies against COX-2 revealed that it displayed approximately about 16- and 5-times stronger interactions than the free Ibu and CCC-Nap ligands, respectively. Moreover, it improved the production of reactive oxygen species (ROS) by 10.7-fold compared to H_2_O_2_, when used as a positive control in MCF-7 cells.

Rana et al., 2021, [74] described the synthesis and study of two pyridine and pyrimidine Ru(II)-NHC complexes, functionalized annulated NHC **3** and **4** with a half-sandwich geometry, and their in vitro cytotoxicity studies against lung (A549), colon (HCT-116) and breast (MCF-7) cancer cell lines and non-cancerous 3T3 cells (embryonic fibroblast isolated from a mouse), using cisplatin as the reference (IC_50_ were 64; 23.2; 14 and 64 µM, respectively) via the MTT assay. Both compounds were more active than the reference, with the exception of **3** against HCT-116 cancer cells, and, overall, compound **4** was more active than compound **3**. Moreover, in silico studies, predicting the binding-sites and atomic interactions of lead molecules with B cell CLL/lymphoma (BCL-2) (PDB entry: 4lvt) and DNA dodecamer (PDB entry: 1bna), suggested that the order of reactivity of the molecules is **4** > **3**. Subsequently, the same group [75] investigated a pyrimidine functionalized non-annulated half-sandwich Ru(II)-NHC complex, namely chloro(*p*-cymene)-1-methyl-3-pyrimidylimidazolideneruthenium(II)-hexafluorophosphate **5**, derived from the non-annulated NHC precursor 1-methyl-3-pyrimidylimidazolium-hexafluorophosphate. The half-sandwich geometry of the molecule was established with single crystal X-ray diffraction. As well, this compound was more active than cisplatin against the cell lines used [74]. Particularly, **5** and **4** were 3- to 4-fold more active than **3** and cisplatin. Predicting binding affinities for **5** and **4** were −7.1 and −7.0 kcal/mol^–1^, respectively, for BCL-2 and −7.3 and −8.2 kcal/mol^–1^, respectively, for DNA. DNA cleavage activity of complex **5** confirmed the ability of ruthenium to perform a direct double-strand breaking. Moreover, molecular docking analysis suggested that complex **5** binds, with the highest binding affinity, a hydrophobic pocket in BLC-2, different from that of complexes **3** and **4**. The opposite was found in the contact simulation of the complexes with a DNA strand: complexes **4** and **5** are superimposed, while complex **3** binds the other side of the DNA strand. However, the binding affinity predicted for **4** was higher than **5**.

Rodriguez-Prieto et al., 2021, [76] reported the synthesis of three spherical carbosilane metallodendrimers of different generations holding Ru-NHC complexes. Compounds **6** and **7**, which belong to the first- and second-generation dendrimers, respectively, are shown in Table 1, while the third-generation dendrimer is not shown. These compounds showed cytotoxic activity similar, or even better, than cisplatin against four cancer cell lines, namely advanced prostate (PC3), breast (HCC1806), cervix (HeLa), human liver (HEPG2) and the non-tumoral fibroblast (HFF-1) cell line, as determined by MTT assay. IC_50_ for cisplatin was referred to data of the literature: IC_50_ = 30.18 ± 2.58 µM (MCF-7), 11.75 ± 1.23 µM (HeLa), 17.29 ± 1.05 µM (HFF-1) [77]. The complexes have been proposed as possible candidates for cancer treatment, due to their combined double action, i.e., antitumoral and carrier for anticancer siRNA. These compounds were capable of forming dendriplexes by promoting the entrance of Mcl-1-FITC (myeloid cell leukemia-1 fluorescein labelled) small interfering RNA (siRNA) to HEPG2 cancer cells, protecting the siRNA from RNAse. Moreover, the cellular uptake of the three complexes was studied by confocal microscopy with Mcl-1-FITC siRNA. Particularly, the second-generation dendrimer **7** was more active than first-generation dendrimer **6**. Compound **7** displayed promising antitumoral properties, being selective, even more than cisplatin, against cancer cell lines, with respect to the normal ones. The different activity was related to the inability of first-generation dendrimer **6** to interact with the siRNA. The compounds were internalized into the cells by endocytosis and internalization increased by generation of the dendritic system.

Paşahan et al., 2022, [78] recently reported a study on Ru-NHC complexes as potential anticancer agents, examining their antiproliferative activity against rat glioblastoma (C6) and human cervix adenocarcinoma (HeLa) cell lines by ELISA assay. Four complexes, namely **8–11,** were more active than cisplatin, used as the standard (IC_50_ = 136 ± 0.7 mM (C6) and 126 ± 0.6 mM (HeLa)).

Chen et al., 2020, [79] reported the synthesis and in vitro antiproliferative activity evaluation of a small panel of NHC-coordinated ruthenium(II) arene complexes. The compounds showed cytotoxic activities against the human ovarian A2780 cancer cells. The highest cytotoxic activities were found for **12** and **13**, which were about 2-fold more potent than cisplatin. Furthermore, these compounds induced apoptosis in a caspase-dependent manner, primarily through intracellular ROS overproduction and cell cycle arrest at the G1 phase. Moreover, in a preclinical metastatic model of A2780 tumor xenograft, administration of **12** and **13** resulted in a marked inhibition of tumor progression and metastasis. A significant alleviated systemic toxicity was observed in animals for both complexes in comparison with cisplatin.

Sari et al., [80] designed four (NHC)Ru(II)(*η*^6^-*p*-cymene) complexes, bearing 2-morpholinoethyl and 4-vinylbenzyl substituents to the benzimidazole core and different alkyl/aryl groups to the second nitrogen atom, and studied their cytotoxic activity against MCF-7 breast cancer cells and their DNA-binding properties. The authors individuated the compound **14** as lead, showing an IC_50_ of 3.61 µM, and that it can bind the plasmidic DNA without exerting genotoxic effects.

A recent interesting article by Wilke et al., [81] described the study of the ruthenium complex HB324 (**15**), which showed promising potential as a novel anticancer agent in vitro. This complex showed good effects, even in low micromolar concentrations, especially regarding proliferation inhibition and apoptosis induction via the mitochondrial pathway on human B-cell precursor leukemia Nalm-6 cells. Moreover, of particular interest is the upregulation of the Harakiri resistance protein, which inhibits the anti-apoptotic and death repressor proteins BCL-2 and BCL-xL. Finally, compound **15** showed synergistic activity with various established anticancer drugs, including vincristine, and overcame the resistance in several cell lines, such as neuroblastoma cells.

### 3.2. Antimicrobial Ru(II)-NHC Complexes Possessing Other Additional Abilities (Antiproliferative, Antioxidant and Anticholinesterase)

Antimicrobic resistance is a cause of great concern worldwide and significantly affects humanity’s capacity to prevent and treat a growing number of bacterial and fungal infections [82]. In the past decade, the antimicrobial activity of ruthenium complexes has been reviewed [83] and studied as antimicrobial agents and alternatives or adjuvants to the more traditional antibiotics [84]. The recent antimicrobial studies regarding complexes of NHC-ruthenium are summarized in Table 2, and the minimal inhibitory concentrations (MICs) are reported.

Roymahapatra et al., 2015, [85] described the synthesis, pro-apoptotic and antimicrobial studies of two pyrazine functionalized Ru(II)-NHC complexes of methylimidazolylidene and methylbenzimidazolylidene, namely bis-[2,6-di-(*N*-methylimidazol-2-ylidene)pyrazine]ruthenium(II) hexafluorophosphate (**16**) and bis-[2,6-di-(*N*-methylbenzimidazol-2-ylidene)pyrazine]ruthenium(II) hexafluorophosphate (**12**). Inhibition of cell proliferation was studied against human colon carcinoma cell lines (HCT15) and human epidermoid cancer cells (Hep2) by the standard MTT assay. Complex **16** was more active than complex **17** against both cancer cell lines. DNA binding and cleavage studies demonstrated that these complexes may induce cancer cell apoptosis. Moreover, complex **16** showed antimicrobial activities against Gram-positive (*Staphylococcus epidermidis* NCIM2493) and Gram-negative bacteria (*Pseudomonas aeruginosa* ATCC 27853), and antimycotic properties against *Candida albicans* SJ11 (unicellular fungus) by targeting their cell wall and DNA or plasmid inside the cell.

Streciwilk et al., 2018, [86] studied antimicrobial activities of three Ru(II) naphthalimide-NHC complexes against Gram-positive (*Bacillus subtilis* 168 DSM402; *Staphylococcus aureus* DSM 20231 and ATCC 43300) and Gram-negative (*Escherichia coli* DSM 30083, *Acinetobacter baumannii* DSM 30007 and *P. aeruginosa* DSM 50071) bacteria and the cytotoxic activity against MCF-7 human breast cancer and HT-29 colon adenocarcinoma cells. Complexes **18** and **19** showed activity only against Gram-positive bacteria, while showing no activity against Gram-negative (MIC = 256 µg/mL; 366 µM). Compound **19** also showed cytotoxicity against the two cell lines used. The results were compared to cisplatin or 5-fluorouracil (IC_50_ values for standards below 10 µM). DNA-binding studies were also carried out by recording the circular dichroism of B-DNA and c-Kit2 G4 in the presence of increasing amounts of **19**, highlighting a strong binding of **19** to the B-DNA model and suggesting intercalation as a mode of connection. Further studies carried out by the same research group [87] with complex **19** on HCT-116 colorectal cancer cells demonstrated that it triggers apoptosis via a rapid and consistent activation of the ROS-p38 MAPK pathway. Following this, the same group [88] reported a research, similar to the previous one, regarding the antimicrobial and antiproliferative activity of a bis-naphthalimide NHC–ruthenium (II) complex (**20**). It showed a slight antimicrobial effect against Gram-positive bacteria, whereas it was not cytotoxic against the cell lines studied.

Boubakri et al., 2019, [89] studied a series of Ru(II)-NHC complexes against three Gram-positive bacteria (*Micrococcus luteus* LB14110, *Listeria monocytogenes* ATCC19117, *S. aureus* ATCC6538) and three Gram-negative bacteria (*Salmonella typhimurium* ATCC14028, *P. aeruginosa* ATCC 9189, *E. coli*) using the well diffusion method. The reference antibacterial drug used was ampicillin. The complex **21** and **22** were the most active against *M. luteus*, *L. monocytogenes* and *S. typhimurium* (ampicillin, MIC = 0.004, 0.002 and 0.625 mg/mL, respectively). The compound **22** was also active versus *E. coli* (for this compound only an inhibition zone value was given). Moreover, the antifungal activity of the Ru(II)–NHC complexes, with respect to *C. albicans*, was tested via the solid medium diffusion method. Also in this case, the compound **21** was active, showing the highest diameter of growth inhibition (34 ± 0.18 mm). Interestingly, the compound **21** also showed acetylcholinesterase inhibitory activity (AChEI), with a percentage inhibition to the order of 47.1%. Regarding the anticancer activity, the compounds **21** and **22** also showed cytotoxicity against two human breast cancer cell lines, namely MCF-7 and MDA-MB-231, as assessed by MTT assay. The compound **21** also showed antioxidant activity, as demonstrated via the 2,2-diphenyl-1-picrylhydrazyl (DPPH) and 2,2′-azinobis-3-ethylbenzothiazoline-6-sulphonic acid (ABTS) assays, from a concentration of 1 mg/mL, showing a scavenging activity very similar to that of the two controls, i.e., gallic acid and butylated hydroxytoluene. A recent study by the same group [90] deepened the study of compound **21** and highlighted the interesting results obtained for this compound, along with its congener **23.** The compound **21** showed antibacterial activity against *L. monocytogenes*, *S. aureus* and *S. typhimurium*, sometimes equal to, or higher than, references ampicillin (MIC = 3.9 µg/mL, 1.95 µg/mL, 3.9 µg/mL) or kanamycin (MIC = 12.5 µg/mL, 6,25 µg/mL, 12.5 µg/mL). Both the compounds **21** and **23** showed antifungal activity against *C. albicans*, equal to the reference fluconazole (MIC = 1.25 µg/mL). The two compounds also showed moderate cytotoxicity against MCF-7 and MDA-MB-231 cancer cell lines, as assessed by MTT assay. The compounds **21** and **23** were also studied for inhibition of acetylcholinesterase (AChE) and tyrosinase (TyrE), giving interesting results compared to galantamine (IC_50_ = 0.25 µg/mL against AChE) and kojic acid (IC_50_ = 5.05 µg/mL against TyrE). The compound **23** showed an interesting antioxidant activity, similar to that of the standard butylated hydroxytoluene (BHT: EC_50_ = 31.55 µg/mL, 17.41 µg/mL, 89.55 µg/m), as determined for DPPH, ABTS and β-carotene assays, respectively.

Onar et al., 2019, [91] studied four ruthenium NHC complexes for their antimicrobial, anticancer and DNA-binding activities. Antimicrobial activities of the compounds were tested against one Gram-positive (*B. subtilis* ATCC 21332) and one Gram-negative (*E. coli* ATCC 25922) bacteria and one fungal strain (*C. albicans* ATCC 60193). The studied compounds were slightly more active than cefotaxime, used as a reference (MIC ranging from 100 to 200 µg/mL, compared to 250 µg/mL of the standard drug). Interesting results were obtained regarding cytotoxic activity. The benzimidazole-based ruthenium complexes containing a benzyl group **24** and **25** showed cytotoxicity against human colorectal Caco-2 cancer cell lines comparable to cisplatin, whereas they were non-cytotoxic against non-cancer L-929 cell lines.

Slimani et al., 2020, [92] studied four [RuCl_2_(*p*-cymene)]_2_ complexes. The antibacterial activity was tested against three Gram-positive (*Micrococcus luteus* LB 14110, *S. aureus* ATCC 6538, *Listeria monocytogenes* ATCC 19117), and three Gram-negative (*Salmonella typhimurium* ATCC 14028, *P. aeruginosa* ATCC 49189 and *E. coli*) bacteria, using the well diffusion method. Ampicillin was used as a standard control drug (MIC = 0.004 mg/mL, 0.002 mg/mL and 0.625 mg/mL against *M. luteus* LB 14110; L. *monocytogenes* ATCC 19117 and *S. typhimurium* ATCC 14028, respectively). The compounds **26** and **27** demonstrated effective antibacterial activity against the tested bacteria; particularly, the compound **26** showed excellent activity against *S. typhimurium*, being more active than the reference drug. Moreover, the compound **26**, from a concentration of 1 mg/mL, demonstrated a very similar scavenging activity to that of the two controls, butylated hydroxytoluene (BHT) and gallic acid (GA), as determined via DPPH assay. Finally, both **26** and **27** showed cytotoxic activity against human breast cancer cell lines MCF-7 and MDA-MB-231.

Burmeister et al., 2021, [93] studied a series of benzimidazolium cations and the corresponding organometallics of the type (*p*-cymene)(NHC)Ru(II)Cl_2_ as antibacterials against Gram-positive (*B. subtilis, S. aureus* DSM 20231, *S. aureus* ATCC 43300) and Gram-negative (*E. coli, A. baumannii, P. aeruginosa*) strains in a microtiter plate assay, according to CLSI guidelines, using ciprofloxacin as a reference drug. The complex (1,3-dibenzyl-5-bromo-1*H*-benzimidazol-2-ylidene)-dichlorido-(*η*^6^-*p*-cymene)ruthenium(II) **28** was the most active compound against Gram-positive bacteria, while it showed low activity against Gram-negative bacteria. This compound showed moderate inhibition of bacterial TrxR (*E. coli*). Thus, the inhibition of TrxR is unlikely a major mechanism of the antibacterial activity of the ruthenium NHC complexes.

## 4. Conclusions

NHCs have been recognized as a class of strong donating ligands, which may stabilize diverse metal complexes of catalytic importance. Among these, the NHC ruthenium complexes have been, and still are, a fruitful research, filed for various applications: catalytic, photochemical and biological. This review provides an overview of the most recent studies on ruthenium NHCs complexes with biological activities, focusing on the antimicrobial and antiproliferative properties. Most of these compounds showed interesting antimicrobial activity against Gram-positive bacteria and some are also active against Gram-negative bacteria and fungi. The antiproliferative activity was also demonstrated for several compounds belonging to this class against several types of cancer cell lines.

## Figures and Tables

**Figure 1 antibiotics-12-00365-f001:**
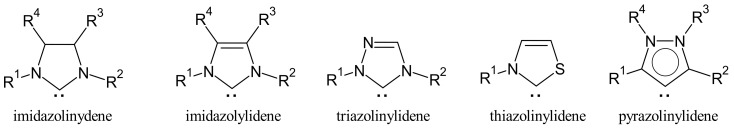
Structures of NHCs.

**Table 1 antibiotics-12-00365-t001:** Antiproliferative activity of Ru(II)-NHC complexes.

Structure	Compd.	Biological Activities	Ref
** 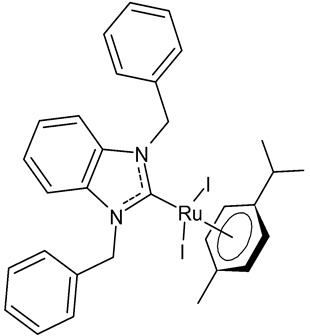 **	**1**	Trx inhibition (%): 71 ± 8 IC_50_ = 6.2 ± 0.4 µM (HCT-116) IC_50_ = 8.4 ± 0.2 µM (SiHa) IC_50_ = 7.8 ± 1.0 µM (NCI-H460)	Lam et al., 2018 [72]
** 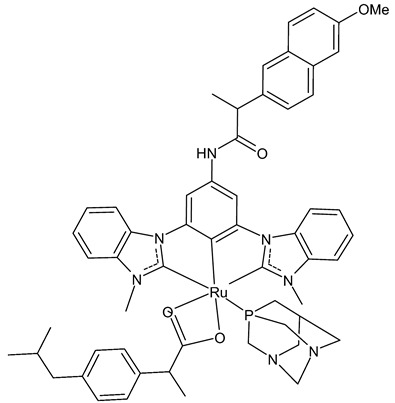 **	**2**	IC_50_ = 0.91 ± 0.02 µM (MCF-7) IC_50_ = 1.32 ± 0.05 µM (MDA-MB-231) IC_50_ = 35.82 ± 0.52 µM (HT-29) IC_50_ = 4.71 ± 0.05 µM (MCF-10A) IC_50_ = 108.20 ± 0.03 µM (HEK-293)	Tabrizi et al., 2019 [73]
** 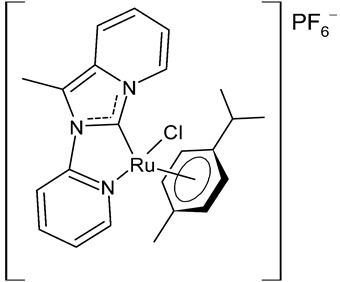 **	**3**	IC_50_ = 28.7 ± 2.3 µM (A549) IC_50_ = > 100 µM (HCT-116) IC_50_ = 14.8 ± 2.3 µM (MCF-7) IC_50_ = 44.64 ± 2.6 µM (3T3)	Rana et al., 2021 [74]
** 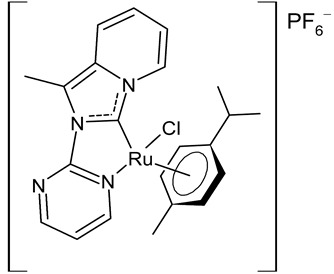 **	**4**	IC_50_ = 2.1 ± 0.7 µM (A549) IC_50_ = 8.6 ± 1.8 µM (HCT-116) IC_50_ = 3.3 ± 0.4 µM (MCF-7) IC_50_ = 9.36 ± 1.16 µM (3T3)	Rana et al., 2021 [74]
** 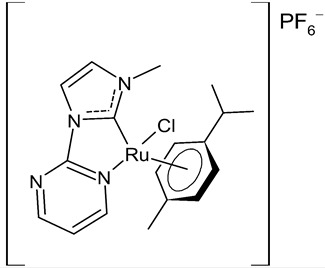 **	**5**	IC_50_ = 2.8 ± 0.4 µM (A549) IC_50_ = 2.3 ± 0.3 µM (HCT-116) IC_50_ = 4.7 ± 0.7 µM (MCF-7) IC_50_ = 8.56 ± 1.6 µM (3T3)	Rana et al., 2020 [75]
** 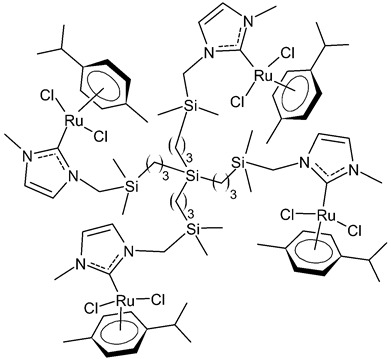 **	**6**	IC_50_ = 37.2 ± 3.6 µM (PC3) IC_50_ = 25.3 ± 7.6 µM (HCC1806) IC_50_ = 71.6 ± 15.4 µM (HeLa) IC_50_ = 10.3 ± 1.7 µM (HEPG2) IC_50_ = 21.2 ± 1.8 µM (HFF-1)	Rodriguez-Prieto et al., 2021 [76]
** 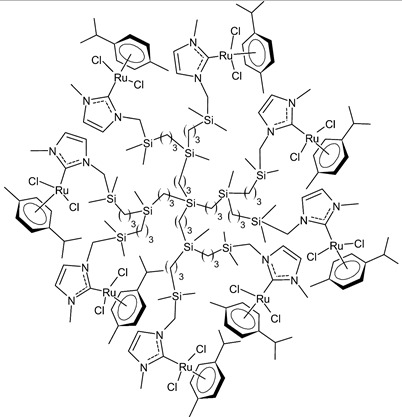 **	**7**	IC_50_ = 21.4 ± 0.9 µM (PC3) IC_50_ = 20.6 ± 1.9 µM (HCC1806) IC_50_ = 8.3 ± 1.1 µM (HeLa) IC_50_ = 6.6 ± 0.5 µM (HEPG2) IC_50_ = 69.3 ± 1.2 µM (HFF-1)	Rodriguez-Prieto et al., 2021 [76]
** 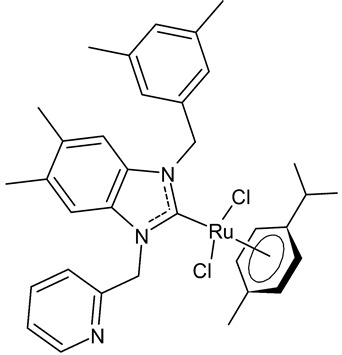 **	**8**	IC_50_ = 14.2 ± 0.5 mM (C6) IC_50_ = 11.1 ± 0.5 mM (HeLa)	Paşahan et al., 2022 [78]
** 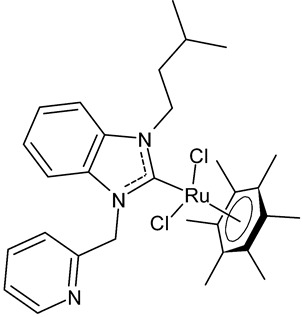 **	**9**	IC_50_ = 16.2 ± 0.4 mM (C6) IC_50_ = 13.7 ± 0.3 mM (HeLa)	Paşahan et al., 2022 [78]
** 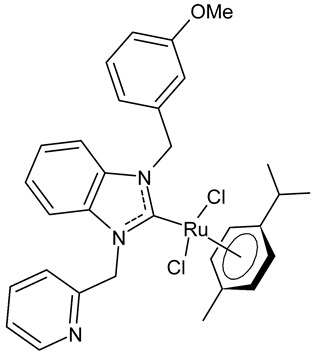 **	**10**	IC_50_ = 24.2 ± 0.7 mM (C6) IC_50_ = 22.8 ± 0.8 mM (HeLa)	Paşahan et al., 2022 [78]
** 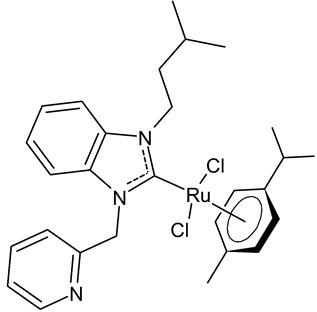 **	**11**	IC_50_ = 37.3 ± 0.9 mM (C6) IC_50_ = 17.3 ± 0.8 mM (HeLa)	Paşahan et al., 2022 [78]
** 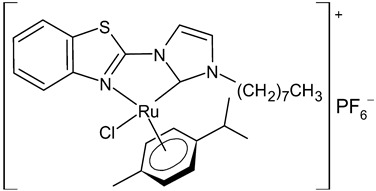 **	**12**	IC_50_ = 2.74 ± 0.15 mM (A2780)	Chen et al., 2020 [79]
** 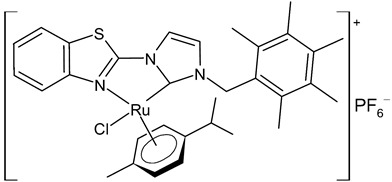 **	**13**	IC_50_ = 1.98 ± 0.10 mM (A2780)	Chen et al., 2020 [79]
** 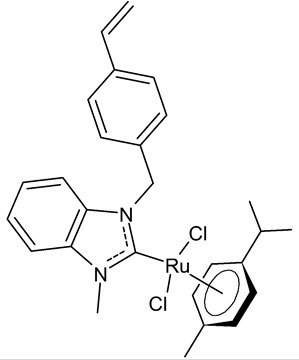 **	**14**	IC_50_ of 3.61 µM (MCF-7)	Sari et al., 2020 [80]
** 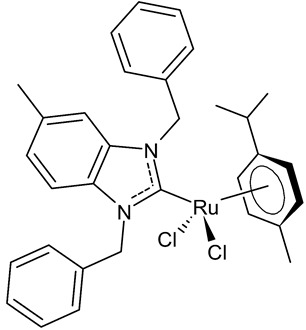 **	**15** (HB324)	G1 Arrest * = ≥1 µM AC50 ** = ~4 µM	Wilke et al., 2023 [81]

* G1 arrest, which means a proliferation inhibition equal to 100% or above, indicating occurring cell death; ** AC_50_ is the concentration necessary to induce apoptosis in half of the cell population.

**Table 2 antibiotics-12-00365-t002:** Antimicrobial and other activities (antiproliferative, antioxidant and anticholinesterase) of Ru(II)-NHC complexes.

Structure	Compd.	Biological Activities	Ref
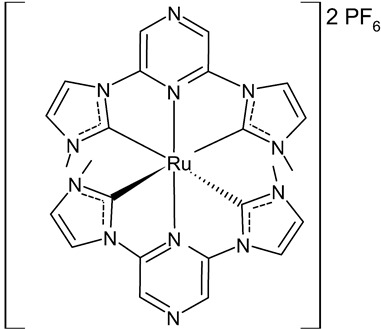	**16**	MIC = 8 µM (*S. epidermidis* NCIM 2493) MIC = 8 µM (*P. aeruginosa* ATCC 27853) MIC = 16 µM (*C. albicans* SJ11, unicellular fungus) IC_50_ = 22.70 ± 1.3 µM (HCT15) _50_ = 18.46 ± 2.3 µM (Hep2)	Roymahapatra et al., 2015 [85]
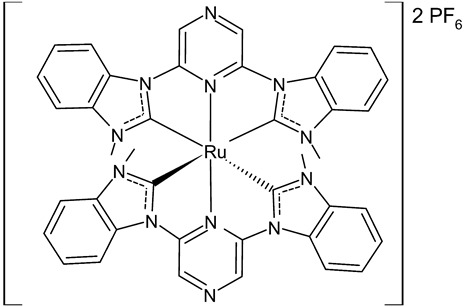	**17**	MIC = 64 µM (*S. epidermidis* NCIM 2493) MIC = 64 µM (*P. aeruginosa* ATCC 27853) MIC = 256 µM (*C. albicans* SJ11, unicellular fungus) IC_50_ = 82.2 ± 4.6 µM (HCT15) IC_50_ = 61.8 ± 3.3 µM (Hep2)	Roymahapatra et al., 2015 [85]
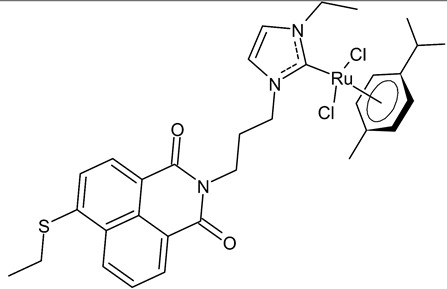	**18**	MIC = 16 µg/mL; 22.9 µM (*B. subtilis* 168 DSM402) MIC = 16 µg/mL; 22.9 µM (*S. aureus* DSM 20231) MIC = 16 µg/mL; 22.9 µM (*S. aureus* ATCC 43300) IC_50_ = 11.6 ± 1.0 µM (MCF-7) IC_50_ = 26.4 ± 1.1 µM (HT-29)	Streciwilk et al., 2018 [86]
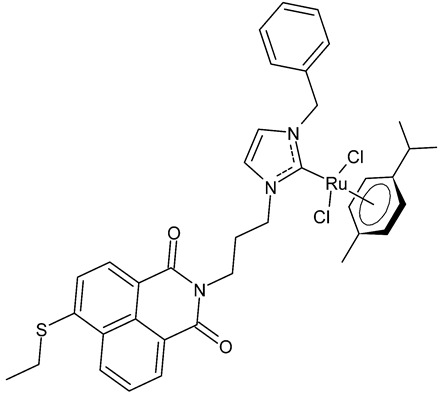	**19**	MIC = 8 µg/mL; 10.5 µM (*B. subtilis* 168 DSM402) MIC = 8 µg/mL; 10.5 µM (*S. aureus* DSM 20231) MIC = 8 µg/mL; 10.5 µM (*S. aureus* ATCC 43300) IC_50_ = 4.8 ± 0.1 µM (MCF-7) IC_50_ = 4.9 ± 0.02 µM (HT-29)	Streciwilk et al., 2018 [86]
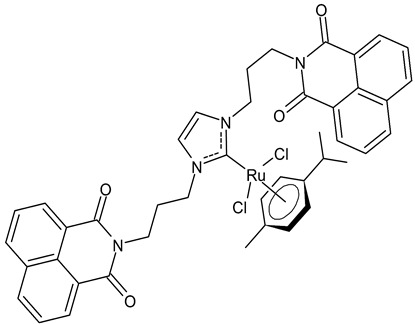	**20**	MIC = 16 µg/mL; 18.8 µM (*B. subtilis* 168 DSM402) MIC = 16 µg/mL; 18.8 µM (*S. aureus* DSM 20231) MIC = 8 µg/mL; 9.4 µM (*S. aureus* ATCC 43300) IC_50_ = 26.0 ± 1.1 µM (MCF-7) IC_50_ > 100 µM (HT-29)	Streciwilk et al., 2018 [88]
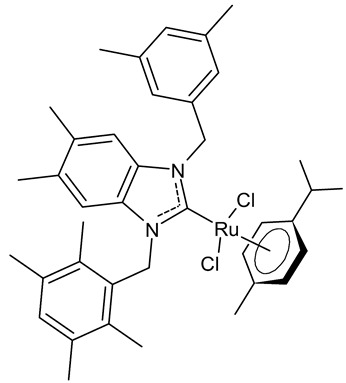	**21**	MIC = 0.0195 mg/mL (*M. luteus* LB14110) MIC = 15.6 µg/mL (*L. monocytogenes* ATCC 19117) MIC = 3.9 µg/mL (*S. aureus* ATCC 6538) MIC = 3.9 µg/mL (*S. typhimurium* ATCC 14028) MIC = 1.25 µg/mL (*C. albicans* ATCC 10231) IC_50_ = 0.67 ± 0.2 µg/mL (MCF-7) IC_50_ = 0.8 ± 0.2 µg/mL (MDA-MB-231) IC_50_ = 2.52 µg/mL (AChE) IC_50_ = 19.88 µg/mL (TyrE)	Boubakri et al., 2019 [89,90]
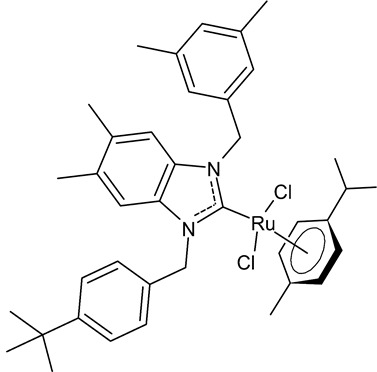	**22**	MIC = 0.0195 mg/mL (*M. luteus* LB14110) MIC = 0.0781 mg/mL (*L. monocytogenes* ATCC 19117) MIC = 1.25 mg/mL (*S. typhimurium* ATCC 14028) Inhibition zone = 15 ± 0.2 mm (*E. coli*) IC_50_ = 0.68 ± 3.2 µg/mL (MCF-7) IC_50_ = 1.93 ± 2.6 µg/mL (MDA-MB-231)	Boubakri et al., 2019 [89]
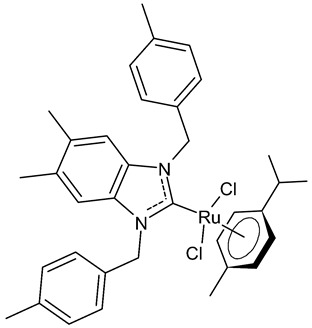	**23**	MIC = 3.9 µg/mL (*L. monocytogenes* ATCC 19117) MIC = 1.95 µg/mL (*S. aureus* ATCC 6538) MIC = 1.95 µg/mL (*S. typhimurium* ATCC 14028) MIC = 1.25 µg/mL (*C. albicans* ATCC 10231) IC_50_ = 0.68 ± 0.2 µg/mL (MCF-7) IC_50_ = 0.8 ± 0.1 µg/mL (MDA-MB-231) IC_50_ = 5.06 µg/mL (AChE) IC_50_ = 24.95 µg/mL (TyrE) EC_50_ = 32.18 µg/mL (DPPH) EC_50_ = 18.17 µg/mL (ABTS) EC_50_ = 92.25 µg/mL (β-carotene)	Boubakri et al., 2022 [90]
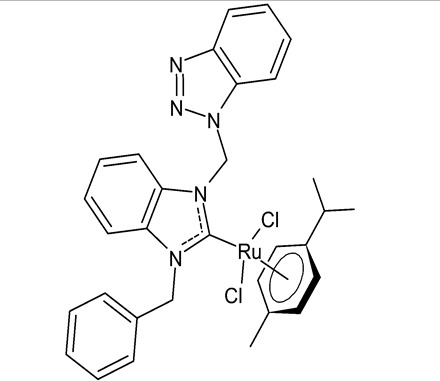	**24**	MIC = 200 µg/mL (*B. subtilis* ATCC 21332) MIC = 100 µg/mL (*E. coli* ATCC 25922) MIC = 200 µg/mL (*C. albicans* ATCC 60193) IC_50_ = 100 ± 7 µM (Caco-2) IC_50_ = 137 ± 2 µM (MCF-7)	Onar et al., 2019 [91]
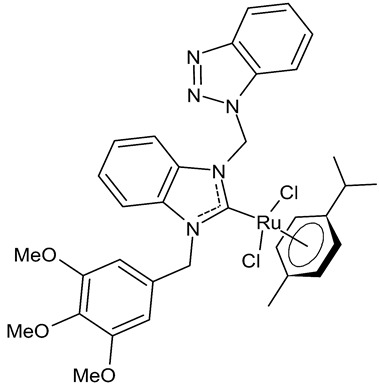	**25**	MIC = 100 µg/mL (*B. subtilis* ATCC 21332) MIC = 200 µg/mL (*E. coli* ATCC 25922) MIC = 200 µg/mL (*C. albicans* ATCC 60193) IC_50_ = 90 ± 1 µM (Caco-2) IC_50_ = 270 ± 12 µM (MCF-7)	Onar et al., 2019 [91]
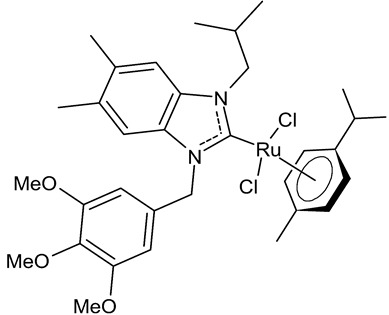	**26**	MIC = 0.0195 mg/mL (*M. luteus* LB 14110) MIC = 0.1562 mg/mL (*L. monocytogenes* ATCC 19117) MIC = 0.0781 mg/mL (*S. typhimurium* ATCC 14028) IC_50_ = 0.6 ± 1.1 µg/mL (MCF-7) IC_50_ = 1.1 ± 0.3 µg/mL (MDA-MB-231)	Slimani et al., 2020 [92]
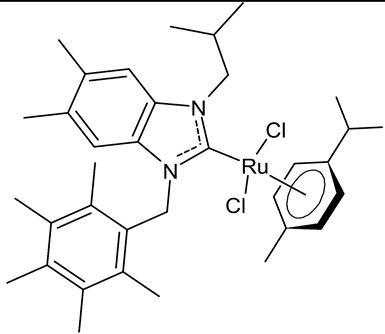	**27**	MIC = 0.0195 mg/mL (*M. luteus* LB 14110) MIC = 0.0781 mg/mL (L*. monocytogenes* ATCC 19117) MIC = 1.25 mg/mL (*S. typhimurium* ATCC 14028) IC_50_ = 0.68 ± 1.2 µg/mL (MCF-7) IC_50_ = 1.7 ± 0.6 µg/mL (MDA-MB-231)	Slimani et al., 2020 [92]
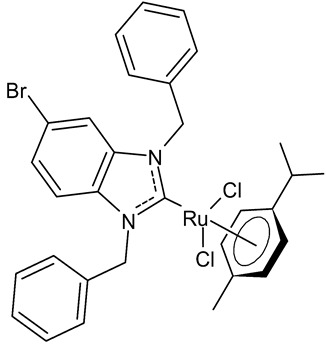	**28**	MIC = 11.7 µM (*B. subtilis*) MIC = 23.4 µM (*S. aureus* DSM 20231) MIC = 11.7 µM (*S. aureus* ATCC 43300)	Burmeister et al., 2021 [93]

## Data Availability

Not applicable.

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
