# Peer review of "Biological Activities of Ruthenium NHC Complexes: An Update"

_antibiotics, 2023, doi:10.3390/antibiotics12020365_

Round 1

Reviewer 1 Report

Catalano et al. describe herein an extensive review of Ru NHC complexes used as inhibitors for their antimicrobal and antiproliferative properties. It is overall a nice summary of the work done in the field by this group and others, and highlights the use of Ru NHC complexes outside of the common use as catalysts in other areas of organic chemistry. Recommend accepting the manuscript for publication in Antibiotics.  

Author Response

We are grateful to reviewer for the comment.

Reviewer 2 Report

This Article reports “Biological Activities of Ruthenium NHC Complexes: An Update” is a significantly important for Biological prospective.

I recommend publishing after a few minor revisions that should be addressed.

(a)    Page 5, Structure 2, The positive charge in Benzimidazole must draw. The ring without charge distribution is not accurate. The sign of charge should be included in imidazole ring, check the structure in cited reference no. 73.

(b)   This correction of missing charge in rings should be included in remaining structure.

(c)    In Table 1 and 2, the authors must insert the title for Sr. No. or Compound No. in first rows of each table.

(d)   In Manuscript description in section 4 (Ruthenium-NHC complexes) the two paragraphs numbering has to change. The first paragraph is missing, or the numbering is wrong, it should be 4.1 not 4.2.

(e)   In reference section: The margin after numbers is different i.e., the numbering format is different, it must change for every reference number and use the journal standard numbering format.

(f)     The cited reference page number must be in a one format (xxx-xxx) not (xxx). Check all cited reference page numbers and correct it.

(g)    In ref. 13, 15, 16, 18, 25, 27 31, 32, and so on.. the author did not keep the space between the cited reference authors initial and middle name. The author must keep the space between initial and middle name of cited authors. Check all cited reference authors name and correct it.

(h)   In ref. 15, the page number is missing.

(i)      In ref. 20, 81, and 43 the published years must in bold format, also correct the volumes and page numbers in ref. 43 and 81.

(j)      In ref. 62, the page of cited reference is missing, use the correct page number format.

Author Response

We are grateful to the reviewer for the useful comments.

Reviewer 3 Report

In this present review, Alessia et al. aimed to provide a succinct, comprehensive review entitled "Biological Activities of Ruthenium NHC Complexes: An Update" in the field of Ru(II)-NHC complexes, which are being studied for their antimicrobial and antiproliferative properties. Overall, the review looks good. The short review is generally well written, but there are a number of errors that should be corrected, and a significant amount of improvement is recommended in the "Ru (II)-NHC complexes with antiproliferative activity" part and a few other portions of the manuscript before further consideration.

Major Revision

Because the authors of this review are attempting to emphasize the importance of the antimicrobial and antiproliferative activities of Ru (II)-NHC complexes, it is not necessary to explain the bioactivity of other metal NHCs (3. metal-NHCs) in this review.

It’s worth describing the potential effect of the second and third generation ruthenium derivative dendrimers, as the first-generation Ru complex is not as potent as the second or third. Include structures for the second and third generations, as well as a few lines of biological activity.

Include the “European Journal of Medicinal Chemistry 203 (2020) 112605” reference in the antiproliferative activity section and discuss.

Include “Journal of Molecular Structure 1202 (2020) 127355” in the reference and discuss.

Include “Int. J. Mol. Sci. 2023, 24(2), 952, https://doi.org/10.3390/ijms24020952” in the reference and discuss.

Page 7, lines 221-224, the authors should consider rewriting the portions for the reference 85 because the MIC for the mentioned compound is misleading.

Page 8, Revisit the ref. 88 (Table 2 of J. Organometal. Chem. 2019, 886, 48-56) and verify the information in the line 264-265 and comment regarding the non-cytotoxicity of the compounds 20 and 21.

Clearly mention which protein PDB ids were used for the docking. Mention whether the binding affinity for 4 was higher than 5 w.r.t. DNA dodecamer and BCL-2.

Minor Revision

Page 1, line 37, "different ligands with the advantage of lower toxicity than platinum complexes"—the line is best supported with the following reference. “Ruthenium Complexes: An Alternative to Platinum Drugs in Colorectal Cancer Treatment, Pharmaceutics 2021, 13(8), 1295”

Mention the range of years covered in this review.

In the line no. 44, it is better to write only "Ruthenium complexes" instead of "Ruthenium (II) complexes" as antivirals for the treatment of COVID-19 because most of the ruthenium compounds examined for COVID-19 are in their +3-oxidation state, even in ref. 22, where the two Ru compounds used for the experiment, NAMI-A and KP-1019, are in their +3-oxidation state. Generally, under certain physiological conditions (e.g., hypoxia or GSH reduction), Ru (III) is reduced to its biologically active form, Ru (II).

On lines 97–99, authors included the line "importance of ruthenium-NHC complexes in organometallic catalysis and bioinorganic chemistry; their significant biological activities have been described as anticancer and antimicrobial agents and referenced as 40," but, in that reference, there is no evidence of Ru-NHC bioactivities. It is suggested that authors put a proper reference in support of the line mentioned in the manuscript.

On lines 121–134, the authors describe the work performed by Tabrizi et al. in 2019. In the work by the Tabrizi, they performed all the assays in vitro. The term "In vitro cytotoxicity" studies must be included in the manuscript.

Add IC50 for complex 2 against all the cell lines in line no. 127 after "Complex 2." It’s easy to understand the comparison with cisplatin.

Page 4, line 144 correct the reference as refs 75 and 76 are not from the same group.

Page 4, line 150, mention the cell lines used.

Page 5, Rana et al. (2021) reference number should be corrected (must be the ref. 74 not 63)

Page 6, the compound number 10 split over two lines. Correct it.

Page 7, line 212-213, use the term “antimicrobial activity” and properly mention the MIC for complexes 13 and 14.

Page 9, Compound 15 structure is incorrect, draw the structure properly with correct substitutions.

Page 9, for compound 14 the reference number should be corrected.

Page 10; reference should be 86 for Boubakri et al. Correct it.

On pages 9 and 10, the compound numbers 16 and 18 have an exactly similar molecular structure. If they are kept intentionally, provide an explanation, and rearrange the numbering of the compounds.

Page 15, Line 520, fix ref 78. The journal name is wrongly abbreviated, and the page numbers are not what they should be.

Author Response

(The authors gave the same response as above.)

Reviewer 4 Report

The review summarized the newly developed Ruthenium N-heterocyclic carbene (NHC) complexes and its anti- microorganism and anti-cancer activity. It’s relevant to the journal’s topic and helpful for medicinal chemists to synthesize better MHC complex. But the manuscript is poorly written, it needs major revision to improve the quality before re-submitting.  

1. The manuscript is not written in a scientific manner, and grammar mistakes could be easily identified throughout the manuscript. The authors should carefully check the manuscript, proofreading or editing service is strongly recommended. For example, line 40 “colorectal cancer treatment is taken”, line 47“N-heterocyclic carbenes (NHCs), the cycloruthenated, and the half sandwiched”, line 56 “which suggest the structural basis of the biomolecular action of these compounds”, line 60 “giving an update of recent studies regarding this relevant topic in medicinal chemistry.’, line 89 “as they demonstrated strong inhibition of the S/ACE2 interaction, particularly of the PLpro enzymatic activity”, line 143the order of reactivity of the molecules was 4>3.”, line 304 “Antiproliferative efficacy was also demonstrated for several compounds belonging to this class against several types of cancer cell lines”.

2. Summary of other’s work should be clearly laid out, highlights and significance of the research should be presented to readers in a logic manner. The author shall try to avoid simply reiterating the published work.

3. The manuscript mostly summarized the chemical structure of the Ruthenium NHC complexes and their antibiotic or anti-proliferation efficacy, more detailed mechanism research should be included. As a review, it’s better to add structure and activity relation (SAR) discussion.

4. The compound from reference 91 should be shown in the table, it would be hard for readers to understand the structure by words. Line 289 “Recently, the synthesis of 2,2’-bipyridyl Ru (II)-annulated NHC complex of 1-methyl-2- pyridin-2-yl-2H-imidazo[1,5-a] pyridin-4-ylidene has been described.”

5. The authors should keep references consistent, half of the references missed the ending page number.

Author Response

We are grateful to reviewer for the useful comments.

Round 2

Reviewer 3 Report

Alessia et al. provide a full review of the antibacterial and antiproliferative activities of Ru NHC complexes. The review seems excellent all around. Although the review is mostly well written, there are a few mistakes that need to be fixed before publication.

Minor Revision

Ø  I would recommend again to check the ref 89 article properly because according to Streciwilk, et al. the result for the compounds is different what is reported here. No need to remove any of the IC50 or MIC as IC50 is the Inhibitory Concentration 50 and MIC represents Minimal Inhibitory Concentration, therefore no need to remove any of the values. Specifically, the MIC is defined as the lowest or minimum antimicrobial concentration that inhibits visible microbial growth in artificial media after a fixed incubation time. And for IC50, at what antibiotic drug concentration is 50% of bacterial growth inhibited.

Ø  I suggested to revisit the ref. 92 (Table 2 of J. Organometal. Chem. 2019, 886, 48-56) because Onar et al. claimed that the two synthesized complexes did not exhibit cytotoxicity against non-cancer L-929 cell lines, despite the fact that it was mentioned in J. Organometal. Chem. 2019, 886, 48-56 in table 2 that those were not determined for cell toxicity for the L-929 cells.

Ø  Check ref. 44 and correct.

Ø  Authors should maintain the uniformity of the ring charge for all the drawn compounds. Check compounds 12 and 13.

Author Response

We thank the referee for the helpful comments.

Answers REVIEWER 3

Ø I would recommend again to check the ref 89 article properly because according to Streciwilk, et al. the result for the compounds is different what is reported here. No need to remove any of the IC50 or MIC as IC50 is the Inhibitory Concentration 50 and MIC represents Minimal Inhibitory Concentration, therefore no need to remove any of the values. Specifically, the MIC is defined as the lowest or minimum antimicrobial concentration that inhibits visible microbial growth in artificial media after a fixed incubation time. And for IC50, at what antibiotic drug concentration is 50% of bacterial growth inhibited.

We thank the referee for this suggestion. The structure of complex 20 and the ref. in table 2 have been corrected.

Ø I suggested to revisit the ref. 92 (Table 2 of J. Organometal. Chem. 2019, 886, 48-56) because Onar et al. claimed that the two synthesized complexes did not exhibit cytotoxicity against non-cancer L-929 cell lines, despite the fact that it was mentioned in J. Organometal. Chem. 2019, 886, 48-56 in table 2 that those were not determined for cell toxicity for the L-929 cells.

We thank the referee for this suggestion. Actually, the IC50 values for the two compounds were reported in the table of ref. 92 (Onar et al.) as N.D. (not determined). It was likely due to the lack of cytotoxicity against this cell line, so the Authors didn’t calculate IC50. Anyway, the words “not given” in the table could be misleading. Thus, we deleted L-929 in the table; in the text the absence of cytotoxicity was already mentioned.

Ø Check ref. 44 and correct

We thank the referee for this correction. It was done.

 Ø Authors should maintain the uniformity of the ring charge for all the drawn compounds. Check compounds 12 and 13.

We thank the referee for this suggestion. The structure of complex 13 was corrected.

Reviewer 4 Report

I’m fine with the revised manuscript, it can be published.

Author Response

We are grateful to the reviewer for the comment.